# Evaluation of Lacustrine Shale Brittleness and Its Controlling Factors: A Case Study from the Jurassic Lianggaoshan Formation, Sichuan Basin

Hongsheng Huang [1], Shuangfang Lu [2,3,*], Pengfei Zhang [4], Qi Zhi [1], Junjie Wang [1] and Zizhi Lin [1]

1   School of Geoscience, China University of Petroleum (East China), Qingdao 266580, China
2   Sanya Offshore Oil & Gas Research Institute, Northeast Petroleum University, Sanya 572025, China
3   Key Laboratory of Continental Shale Hydrocarbon Accumulation and Efficient Development, Ministry of Education, Northeast Petroleum University, Daqing 163318, China
4   College of Earth Science and Engineering, Shandong University of Science and Technology, Qingdao 266590, China
*   Correspondence: lushuangfang@nepu.edu.cn

**Abstract:** To investigate the brittleness of shale and its influencing factors, triaxial rock mechanics experiments, combined with X-ray diffraction, total organic carbon (TOC) measurement, rock pyrolysis, and scanning electron microscopy, were conducted on shales from the Jurassic Lianggaoshan Formation in the Sichuan Basin. $BI_1$, based on the elastic modulus and hardness, $BI_2$, based on mineral composition, $BI_3$, based on strength parameters, and $BI_4$, based on the post-peak energy of shale, were calculated. Additionally, the effects of mineral composition, density, hardness, and organic matter on the brittleness of shales were analyzed. The results show that the shale mineral compositions were dominated by quartz (mean of 45.21%) and clay minerals (mean of 45.04%), with low carbonate mineral contents and high TOC contents. The stress–strain curve showed strong brittleness characteristics. When comparing different evaluation methods, the brittleness evaluation method based on the stress–strain curve (damage energy) was found to be more effective than the mineral fraction and strength parameter methods. The higher the density and hardness, the more brittle the shale. The higher the organic matter and quartz content, the less brittle the shale. The brittleness of sub-member I of the Lianggaosan Formation in Well XQ1 was higher than that of sub-members II and III. This study investigated the brittleness of lacustrine shale and its influencing factors, which is beneficial for the development of shale oil in the Sichuan Basin.

**Keywords:** brittleness; triaxial rock mechanics; lacustrine shale; Lianggaoshan Formation; Sichuan Basin

## 1. Introduction

Shale oil is an essential part of unconventional oil and gas, and the technically recoverable resources of shale oil in China are abundant, with a size of up to $160 \times 108$ t [1]. Recently, a series of breakthroughs have been made in shale oil exploration and development in China, including in the Qijia–Gulong Depression, Changling Depression [2], and Sanzhao Depression in the Songliao Basin; the Jiyang Depression in the Bohai Bay Basin [3,4]; the Jimsar Depression in the Dzungar Basin [5]; and the Gaoyou Sag in the Subei Basin [6].

Shale oil's low porosity and ultra-low permeability impede its ability to flow [7–9]. Unconventional oil and gas accumulations are generally distributed continuously over a large area in the slope or center of a basin, and nanoscale pore throats are widely developed in shale system reservoirs [10]. Zou et al. (2011) discovered oil and gas nanopores smaller than 1 μm for the first time in unconventional oil and gas reservoirs in China [11]. However, shale oil has been effectively developed based on horizontal wells combined with hydraulic fracturing technologies. The brittleness index is the basis for evaluating the fracturability

of shale, and researchers have established many brittleness assessment indices in terms of physical, mechanical, and elastic parameters. However, there is still no unified description of brittleness [12]. As a critical parameter in shale oil evaluation, the brittleness index has an essential impact on the fracturing and low-cost extraction of shale reservoirs [13]. Moreover, the engineering stability analysis, the drillability classification, and disaster prevention and control are primarily controlled by shale brittleness [14]. Rocks with a higher brittleness can form a more extensive scale seam network system after effective fracturing, which can increase shale oil production [15]. Brittleness is considered a composite property used to describe the ability of a rock material to develop into a spatial fracture due to local damage [16–20]. This ability is generally considered to be due to the non-uniform distribution of stresses resulting from the non-uniform distribution of the mineral composition [21]. However, the application of brittleness to characterize comprehensive properties is controversial [21–29].

Brittle rocks are susceptible to sudden damage under external forces, which would produce only small inelastic deformation. According to different definitions, various evaluation methods have been proposed by scholars. Jarvie [21] proposed the mineral fraction method for determining rocks' brittleness and plastic mineral content. Meanwhile, the elastic parameter and complete stress–strain curve characteristic parameter methods based on the stress–strain curves of indoor rock mechanics experiments are generally used [22–24]. Moreover, hardness and strength methods were established according to hardness [25] and strength parameters [26,27] obtained from relevant tests [28].

The brittleness of shale is generally affected by the reservoir's physical properties, mineral composition, organic matter, burial environment, and other factors. It is a primary index for evaluating brittleness [29]. The higher the density and hardness of a rock, the more uniform its internal stress distribution, and the greater its brittleness [25,28]. The effect of mineral composition on rock brittleness is debated. Investigations in North America indicated that quartz primarily positively affected brittleness [21]. However, some experimental results show that the influence of minerals on brittleness is exceptionally complex [30]. Organic matter generally reduces brittleness. Increasing burial depth can make rocks more brittle by increasing their densities [31].

In this study, triaxial rock mechanics tests were performed on shales collected from the Lianggaoshan Formation in the Sichuan Basin. The shale brittleness was studied using four methods (elastic modulus, mineral composition, strength parameter, and stress–strain curve), and the differences in the various evaluation methods were analyzed. This study focused more on the effect of energy change on rock brittleness before and after rock damage by observing the stress–strain curve patterns, and aimed to investigate the brittleness of lacustrine shales and understand its controlling factors. Moreover, the factors influencing shale brittleness are also discussed. This paper provides insight into the brittleness of lacustrine shales, which is beneficial for the effective development of lacustrine shale oil, and contributes to the understanding of global petroleum systems [32–35].

## 2. Geological Setting

The Sichuan Basin is one of the most significant basins in China, being primarily located in Sichuan Province, with an area of approximately 260,000 km$^2$ (Figure 1a). The study area is located to the east of the Longquanshan Deep Fault and west of the Huayinshan Deep Fault (Figure 1b), including the southern part of the Sichuan North Low Flat Tectonic and the Sichuan Central Low Flat Tectonic Zones. The blocky rigid basement in the central Sichuan area leads to upward and downward movements and weaker fold deformation during successive tectonic movements [36,37]. The Lianggaoshan Formation is a set of fine-grained sediments, which can be divided into upper and lower Lianggaoshan members. The upper Lianggaoshan section can be further divided into three sub-members (Figure 1b).

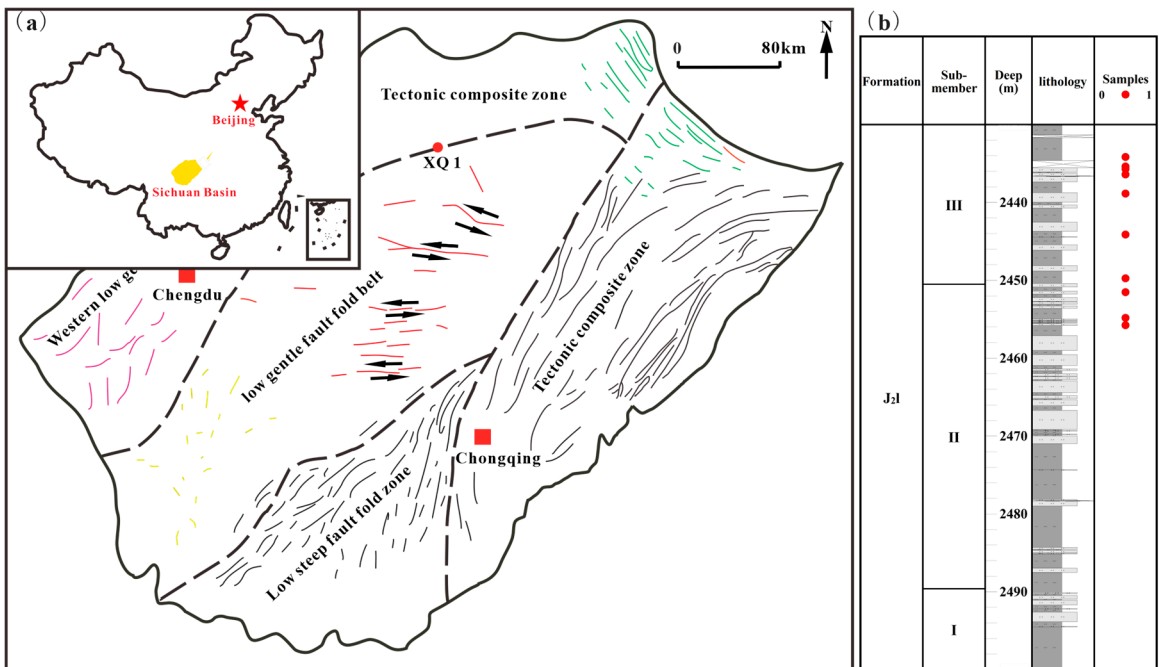

**Figure 1.** Geological Setting: (**a**) Geographical location and tectonic map of the Sichuan Basin. (**b**) XQ1 well logging profile.

The Jurassic system in the Sichuan Basin, covering an area of approximately $18 \times 10^4$ km$^2$, with a stratigraphic thickness of 500~4500 m, is an important oil- and gas-bearing formation [38]. This system has developed several sets of organic-rich black shales, which are important features for shale oil exploration in the Sichuan Basin [39]. Oil and gas exploration and development in the Jurassic system has mainly been concentrated on the central Sichuan area, while seismic studies are mainly focused on the Triassic and deeper layers [38].

## 3. Samples and Methodology

### 3.1. Samples and Experiments

Ten fresh shale core samples were collected from Well XQ1 located in the Yingshan area of central Sichuan Basin, China. All the shale samples were dark gray shales from the Lianggaoshan Formation, collected at a depth ranging from 2430 m to 2470 m. The core samples were cut into cylinders of 50 mm in length and 25 mm in diameter using a wireline cutter.

X-ray diffraction (XRD) analysis, TOC analysis, Rock-Eval pyrolysis, scanning electron microscopy (SEM), and triaxial compression tests were all conducted on the ten samples. The mineral composition was determined by XRD analysis. Shale samples were crushed into powders (<100 mesh), and then the experiment was performed on a TD-3500 X-ray diffractometer [40]. The minerals and the relative mineral percentages for each sample were estimated following the Chinese Oil and Gas Industry Standard SY/T 5163-2010. The TOC content was measured using a CS-230 carbon and sulfur analyzer following the Chinese National Standard GB/T19145-2003. Powder samples (<100 mesh) were analyzed with the Rock-Eval VI to ascertain their thermal maturity. The experiment was carried out according to the Chinese National Standard GB/T 18602-2012, and $S_1$, $S_2$, and other thermal parameters were obtained [41]. Scanning electron microscopy (SEM) is an easily accessible and widely used method for image acquisition and analysis used to determine the morphological characteristics of shale surfaces [40]. The shale samples were ground into small thin sections and then observed with a Phenom ProX scanning electron microscope. The triaxial compression test is an effective method of obtaining the mechanical parameters of shale. The studied samples were loaded into a triaxial pressure tester, and the enclosing

pressure is increased to the theoretical value of the study sample in the formation and deformed axially until the specimen was destroyed. The stress–strain curve of the rock destruction was then recorded, and its various moduli of elasticity were calculated [42].

*3.2. Evaluation Methods for Brittleness*

3.2.1. Evaluation of Rock Brittleness Based on Elastic Modulus and Hardness

Young's modulus and Poisson's ratio are essential parameters in characterizing the brittleness of rocks. Young's modulus is the ratio of stress to strain increments during the elastic deformation phase of a rock, and Poisson's ratio is the ratio of positive longitudinal to positive axial strains. Additionally, rock hardness is the ability of a rock to resist the intrusion of other objects into its surface. It is also used as a critical geological parameter in the design of open-pit blasting, which is vital for shale fracture evaluation [12].

The larger the Young's modulus and hardness of the rock, the smaller the Poisson's ratio, and the more brittle it is. According to the Rickman equation [43] combined with the geological characteristics of the study area, the elastic modulus and Richter hardness were normalized to establish a modified brittleness index model as follows:

$$BI_E = (E_s - E_{smin})/(E_{smax} - E_{smin}), \tag{1}$$

$$BI_v = (v_{max} - v_s)/(E_{smax} - E_{smin}), \tag{2}$$

$$BI_{HL} = (HL - HL_{min})/(HL_{max} - HL_{min}), \tag{3}$$

$$BI_1 = (BI_E + BI_v + BI_{HL})/3, \tag{4}$$

where $Es$ is the static Young's modulus, GPa; $v_s$ is the static Poisson's ratio; $HL$ is the Richter Hardness; $BI_E$ is the normalized Young's modulus; $BIv$ is the normalized Poisson's ratio; $BI_{HL}$ is the normalized Richter hardness; and $BI_1$ is the brittleness index.

3.2.2. Evaluation of Rock Brittleness Based on Mineral Composition

The relative contents of quartz and clay minerals in shale can be determined using XRD analysis. It is generally believed that the relative contents of brittle minerals and clay minerals have an important influence on fracturing. The higher the content of brittle minerals, the easier it is to fracture shale reservoirs. By analyzing the Barnett Shale gas reservoir in North America, Jarvie et al. [21] concluded that the content of quartz in rocks has a significant effect on rock brittleness and proposed using the percentage of quartz in rock minerals to characterize brittleness as follows:

$$BI_2 = \omega(Si)/\{\omega(Si) + \omega(Car) + \omega(Clay)\}, \tag{5}$$

where $\omega(Si)$ is the content of felsic minerals; $\omega(Car)$ is the content of carbonate minerals; $\omega(Clay)$ is the content of clay minerals; and $BI_2$ is the mineral brittleness index.

3.2.3. Evaluation of Rock Brittleness Based on Strength Parameters

Bishop considered the rock rupture characteristics in the post-stress–strain peak phase and proposed the ratio of the difference between the peak and residual strength to the peak strength as the brittleness index, as follows [30]:

$$BI_3 = (\sigma_{1p} - \sigma_{1r})/\sigma_{1p}, \tag{6}$$

where $\sigma_{1p}$ is the peak stress, MPa; $\sigma_{1r}$ is the residual stress, MPa; and $BI_3$ is the strength brittleness index.

3.2.4. Rock Brittleness Evaluation Based on Post-Peak Energy

Tarasov and Potvin characterized rock brittleness by calculating the ratio between the elastic energy absorbed by the rock during triaxial damage and the energy of rupture (or energy released) after the peak of the stress–strain curve, as follows [44]:

$$BI_4 = (M - E)/M, \tag{7}$$

where $BI_4$ is the brittleness evaluation index; $M$ is the post-peak modulus; and $E$ is the elastic modulus. The larger the value of $BI_4$, the more brittle the rock is.

## 4. Results and Discussion

### 4.1. Organic Characteristics and Mineral Compositions

The TOC contents of the selected shales ranged from 0.54 wt.% to 3.27 wt.%, with a mean of 1.64 wt.%. The $S_1$ content varied from 0.01 mg/g to 8.46 mg/g (mean of 1.08 mg/g). $S_2$ values ranged from 0.35 mg/g to 17.25 mg/g, with a mean of 5.82 mg/g. The average Tmax was 447 °C, ranging from 444 °C to 450 °C, and the HI index ranged from 108.63 to 528.17, with a mean of 293.85 mg/g. The TOC values were relatively stable and had a significant correlation with S1 (Table 1). This indicates that the studied samples were developed mainly in a semi-deep lacustrine–deep lacustrine sedimentary environment. Their organic matter type was dominated by types I and $II_1$, with high organic matter maturity, which is favorable for oil production [45–47].

**Table 1.** Organic geochemical characteristics of selected shale samples.

| Sample | Depth/m | Tmax/°C | $S_1$/(mg·g$^{-1}$) | $S_2$/(mg·g$^{-1}$) | TOC/wt.% | HI |
|---|---|---|---|---|---|---|
| S1 | 2434.17 | 447 | 0.37 | 1.37 | 0.93 | 147.80 |
| S2 | 2435.38 | 449 | 0.09 | 0.36 | 0.54 | 108.63 |
| S3 | 2435.71 | 450 | 0.11 | 0.46 | 0.62 | 131.06 |
| S4 | 2436.41 | 446 | 2.83 | 11.38 | 2.69 | 504.41 |
| S5 | 2438.86 | 446 | 4.13 | 17.25 | 3.27 | 528.17 |
| S6 | 2444.13 | 447 | 1.73 | 7.89 | 1.98 | 398.28 |
| S7 | 2449.74 | 448 | 0.12 | 0.35 | 0.64 | 115.10 |
| S8 | 2451.51 | 447 | 0.92 | 5.27 | 1.61 | 326.52 |
| S9 | 2454.82 | 444 | 1.25 | 5.58 | 1.65 | 339.21 |
| S10 | 2455.76 | 447 | 1.46 | 8.26 | 2.43 | 339.36 |

The Barnett Shale in North America contains 20% to 80% quartz, feldspar, and pyrite (from 40% to 60% quartz), less than 25% carbonate minerals, and usually less than 50% clay minerals [21,43]. The studied shales were mainly composed of quartz and clay minerals, with average contents of 45.21% (33.54–55.5%) and 45.04% (37.4–59.25%), respectively, but had low contents of carbonate minerals (Table 2). Compared with the Barnett Shale, the studied shales had a similar silica content and a high carbonate mineral content.

**Table 2.** Mineral compositions of selected shale samples.

| Sample | Clay/% | Quartz/% | Orthoclase/% | Feldspar/% | Calcite/% | Dolomite/% | Pyrite/% |
|---|---|---|---|---|---|---|---|
| S1 | 47.79 | 41.72 | 0.00 | 9.27 | 0.61 | 0.00 | 0.00 |
| S2 | 49.10 | 39.58 | 0.00 | 9.84 | 0.77 | 0.00 | 0.00 |
| S3 | 54.06 | 36.52 | 0.00 | 8.37 | 0.00 | 0.00 | 0.00 |
| S4 | 59.25 | 33.54 | 0.00 | 4.96 | 0.65 | 0.00 | 0.86 |
| S5 | 37.40 | 55.50 | 0.70 | 3.20 | 1.00 | 0.00 | 2.20 |
| S6 | 42.50 | 42.80 | 1.30 | 13.20 | 0.00 | 0.00 | 0.20 |
| S7 | 44.80 | 44.10 | 1.00 | 5.60 | 2.20 | 2.00 | 0.30 |
| S8 | 50.50 | 42.20 | 1.10 | 3.90 | 1.20 | 0.00 | 1.10 |
| S9 | 45.60 | 41.30 | 0.20 | 7.50 | 0.70 | 0.00 | 4.70 |
| S10 | 47.50 | 44.10 | 0.90 | 5.90 | 0.00 | 0.00 | 1.60 |

### 4.2. Pore Types, Porosity, and Permeability of Shales

The rock densities of the selected samples ranged from 2.42 g/cm$^3$ to 2.69 g/cm$^3$, with an average of 2.58 g/cm$^3$. Their porosity varied from 0.41% to 5.79%, with a mean of 2.60%, while the average permeability was 0.367 mD (0.002–2.116 mD) (Table 3). This indicates that the studied shales were typical tight reservoirs, characterized by low porosity and permeability, but large density [48,49]. SEM images show that the pores of the studied shales were dominated by intraparticle pores in clay minerals, followed by interparticle pores at quartz edges and microfractures (Figure 2).

**Table 3.** Physical characteristics of selected shale samples.

| Sample | Depth/m | Porosity/% | Permeability/mD | Density/cm$^3$ |
|--------|---------|-----------|-----------------|----------------|
| S1 | 2434.17 | 1.89 | 0.004 | 2.679 |
| S2 | 2435.38 | 4.65 | 0.244 | 2.567 |
| S3 | 2435.71 | 2.68 | 0.567 | 2.616 |
| S4 | 2436.41 | 4.89 | 0.463 | 2.454 |
| S5 | 2438.86 | 5.79 | 0.129 | 2.419 |
| S6 | 2444.13 | 4.58 | 0.845 | 2.505 |
| S7 | 2449.74 | 2.16 | 0.498 | 2.630 |
| S8 | 2451.51 | 1.49 | 0.100 | 2.594 |
| S9 | 2454.82 | 1.35 | 1.036 | 2.578 |
| S10 | 2455.76 | 2.08 | 0.465 | 2.559 |

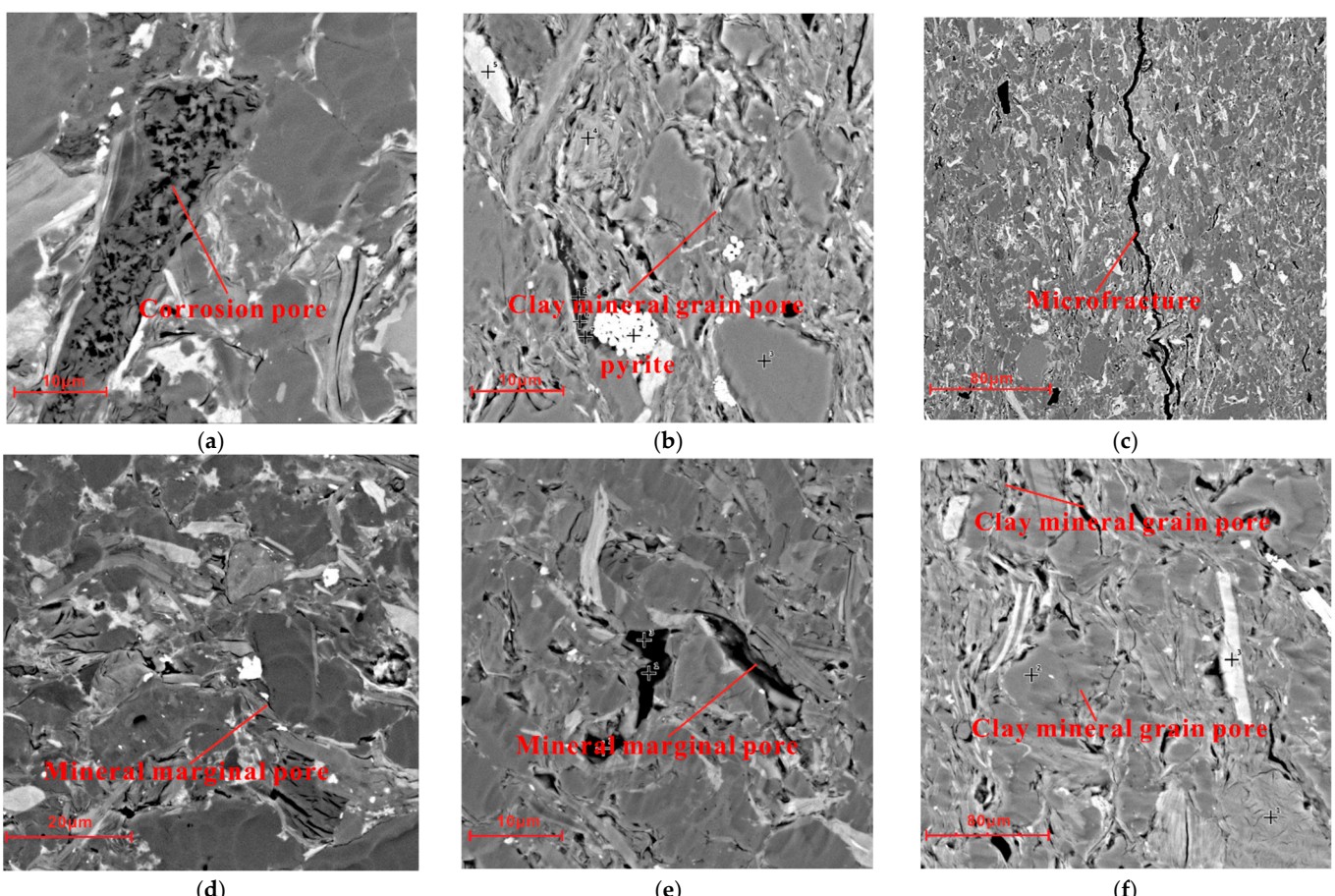

**Figure 2.** SEM images of the studied shales: (**a**) Corrosion pore. (**b**) Clay mineral grain pore. (**c**) Microfracture. (**d**) Mineral grain pore. (**e**) Mineral grain pore. (**f**) Clay mineral grain pore.

*4.3. Shale Brittleness*

4.3.1. Stress–Strain Curves of Shales

Different shale samples had significantly different deformation characteristics under circumferential pressure (36 MPa). The stress–strain curves of the samples mainly showed elastic–plastic deformation, and the damage mode of the test samples was mainly shear damage. Moreover, the samples still had some residual strength after the peak damage. The stress–strain curve patterns of the selected shales can be divided into two categories. The peak stress of organic-lean shales was approximately 190 MPa, and the rock was mainly deformed elastically before rupture, with almost no plastic deformation (Figure 3a). The peak stress of organic-rich shales was approximately 130 MPa, and the rock displayed an obvious plastic deformation stage before rupture (Figure 3b).

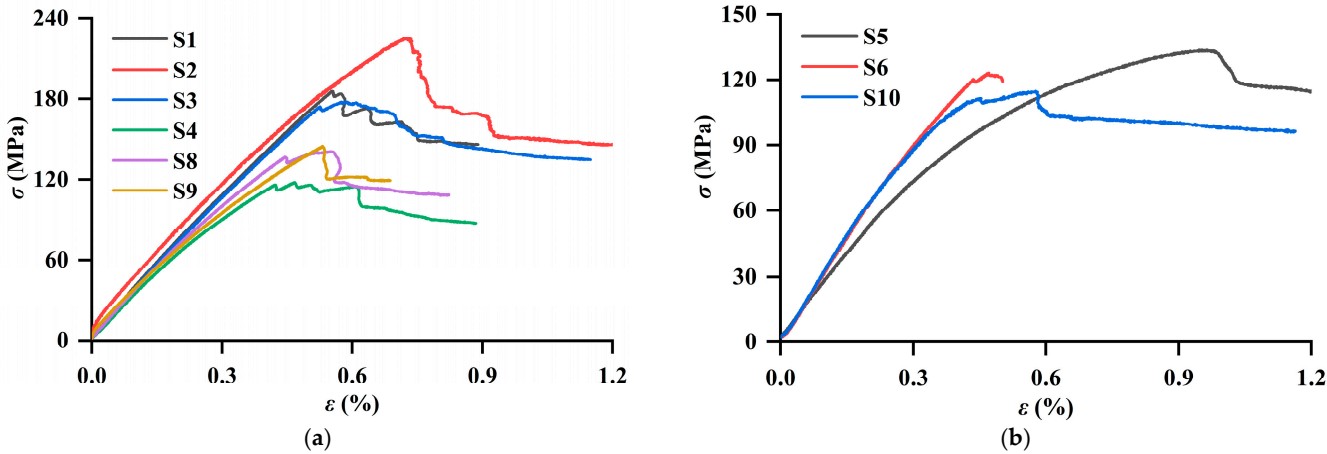

**Figure 3.** Stress–strain curves of different shales: (**a**) Stress–strain curves of organic-lean shales. (**b**) Stress–strain curves of organic-rich shales.

4.3.2. Rock Mechanical Parameters

The rock mechanical parameters of shales with different TOC contents varied. The partial breaking stress of the organic-rich shales ranged between 114.8 MPa and 145.2 MPa (mean of 129.43 MPa). The residual stress ranged from 87 MPa to 120 MPa, with an average of 107.33 MPa, and the average compressive strength was 165.4 MPa (150.8–181.2 MPa). Moreover, Young's modulus ranged from 22.96 GPa to 32.73 GPa, with a mean of 28.96 GPa, while Poisson's ratio distribution ranged from 0.219 to 0.316 (mean of 0.265) (Table 4).

**Table 4.** Rock mechanical parameters of selected shale samples.

| Sample | Failure Deviatoric Stress/MPa | Residual Stress/MPa | Compressive Strength/MPa | Young's Modulus/GPa | Poisson's Ratio |
|---|---|---|---|---|---|
| S1 | 186.1 | 147 | 222.1 | 33.06 | 0.251 |
| S2 | 225.2 | 147 | 261.2 | 32.67 | 0.251 |
| S3 | 178.2 | 135 | 214.2 | 32.72 | 0.254 |
| S4 | 118 | 87 | 154 | 28.59 | 0.228 |
| S5 | 133.8 | 114 | 169.8 | 19.4 | 0.227 |
| S6 | 123.4 | 120 | 159.4 | 29.24 | 0.24 |
| S7 | 179.2 | 164 | 215.2 | 36.86 | 0.249 |
| S8 | 141.4 | 108 | 177.4 | 30.4 | 0.266 |
| S9 | 145.2 | 119 | 181.2 | 27.1 | 0.241 |
| S10 | 114.8 | 96 | 150.8 | 28.25 | 0.237 |

The partial breaking stress of organic-lean shale was 178.2 MPa to 225.2 MPa (mean of 192.18 MPa). The residual stress ranged from 135–164 MPa, with an average of 148.25 MPa, and the average compressive strength was 228.2 MPa (214.2 MPa to 261.2 MPa). Moreover, Young's modulus ranged from 34.02 GPa to 36.31 GPa, with a mean of 34.85 GPa, while Poisson's ratio distribution ranged from 0.248 to 0.305 (mean of 0.248) (Table 4). Organic-lean shales are easier to fracture than organic-rich shales due to their higher modulus of elasticity and other rock mechanical parameters. Compared with organic-rich shales, the various rock mechanical parameters of organic-lean shales have more favorable development.

### 4.4. Brittleness

The brittleness indices calculated using the four methods are listed in Table 5. $BI_1$ ranged from 0.081 to 0.596, with a mean of 0.358, and $BI_2$ varied from 0.175 to 0.731 (mean of 0.431). $BI_3$ ranged from 0.394 to 0.614, with a mean value of 0.499. The average value of $BI_4$ was 0.19, ranging from 0.028 to 0.347. $BI_4$ correlated well with $BI_1$, while the brittleness indices calculated using mineral fraction and mechanical strength ($BI_2$ and $BI_3$) were significantly different from those calculated using other methods (Figure 4). $BI_2$ relates to the mineral compositions, while $BI_3$ was determined using the mechanical parameters before the strain curve. $BI_4$ considers the energy changes during the complete process of rock rupture. Thus, in this study, $BI_4$ is more relevant.

**Table 5.** Calculation of the brittleness index of selected shale samples from elastic modulus and hardness, mineral composition, strength parameters, and post-peak energy ($BI_1$, $BI_2$, $BI_3$, and $BI_4$).

| Samples | $BI_1$ | $BI_2$ | $BI_3$ | $BI_4$ |
|---------|--------|--------|--------|--------|
| S1 | 0.437 | 0.600 | 0.516 | 0.210 |
| S2 | 0.463 | 0.603 | 0.502 | 0.347 |
| S3 | 0.443 | 0.554 | 0.454 | 0.242 |
| S4 | 0.263 | 0.175 | 0.394 | 0.263 |
| S5 | 0.081 | 0.227 | 0.614 | 0.148 |
| S6 | 0.344 | 0.462 | 0.574 | 0.028 |
| S7 | 0.596 | 0.731 | 0.421 | 0.085 |
| S8 | 0.359 | 0.360 | 0.483 | 0.236 |
| S9 | 0.279 | 0.290 | 0.518 | 0.180 |
| S10 | 0.314 | 0.310 | 0.517 | 0.164 |

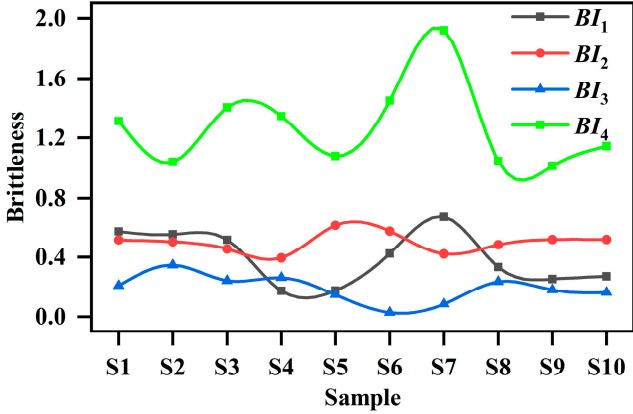

**Figure 4.** Brittleness index of selected shale samples.

### 4.5. Influencing Factors of Brittleness

#### 4.5.1. Effects of Hardness and Density on Brittleness

Figure 5a,b show the relationships between $BI_4$, hardness, and density. $BI_4$ showed significant positive correlations with hardness ($R^2 = 0.8633$) and density ($R^2 = 0.826$), indicating that the denser the shale reservoir, the more difficult it is to fracture the sample in

the triaxial stress test, and the greater the brittleness index calculated from its curve shape. Hence, the rock's brittleness increases as the density and hardness of the rock increase.

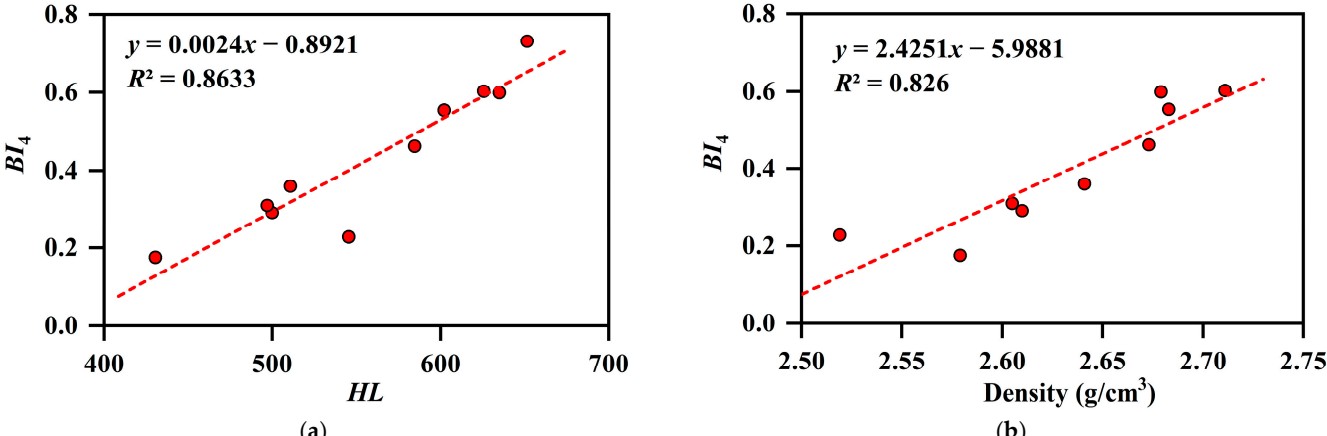

**Figure 5.** Effects of shale hardness and density on brittleness: (**a**) Effects of shale hardness on brittleness. (**b**) Effects of shale density on brittleness.

### 4.5.2. Effects of Organic Matter on Brittleness

Figure 6a,b show the relationships between $BI_4$, $S_1$, and TOC. There were significant negative correlations between the $BI_4$, $S_1$ ($R^2 = 0.7215$), and TOC ($R^2 = 0.7953$) of shales, indicating that shales with high TOC have higher toughness. This is due to organic-rich shales being subject to compaction, and so the organic pores inside the shale can be easily compressed and closed, thus reducing the brittleness. Inorganic pores in shales with a low TOC content are not easily compacted and completed by the rigid mineral lattice, which can enhance the shale's brittleness [26].

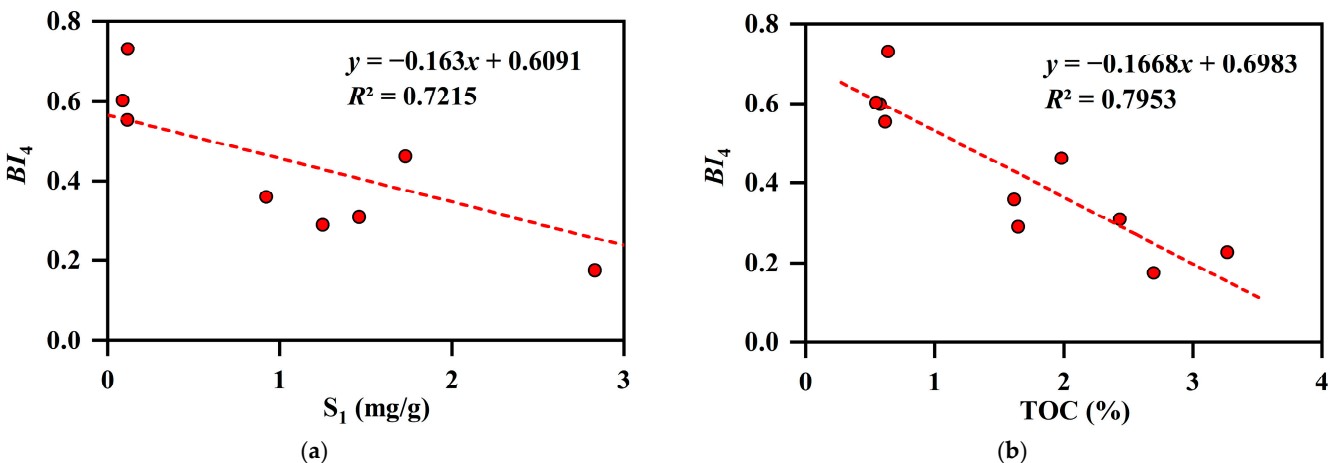

**Figure 6.** Effects of shale $S_1$ and TOC on brittleness: (**a**) Effects of $S_1$ on brittleness. (**b**) Effects of TOC on brittleness.

### 4.5.3. Effects of Mineral Compositions on Brittleness

Figure 7a,b show the relationships between $BI_4$, quartz, and clay mineral contents. $BI_4$ showed a negative correlation with quartz content ($R^2 = 0.5819$) but a significant positive correlation with clay minerals ($R^2 = 0.5331$), indicating that clastic quartz in shales has little or no effect on brittleness [50]. Previous studies show that quartz can affect rock brittleness [21,22,50]. The higher the quartz content of the experimental samples, the lower Young's modulus and the lower the brittleness.

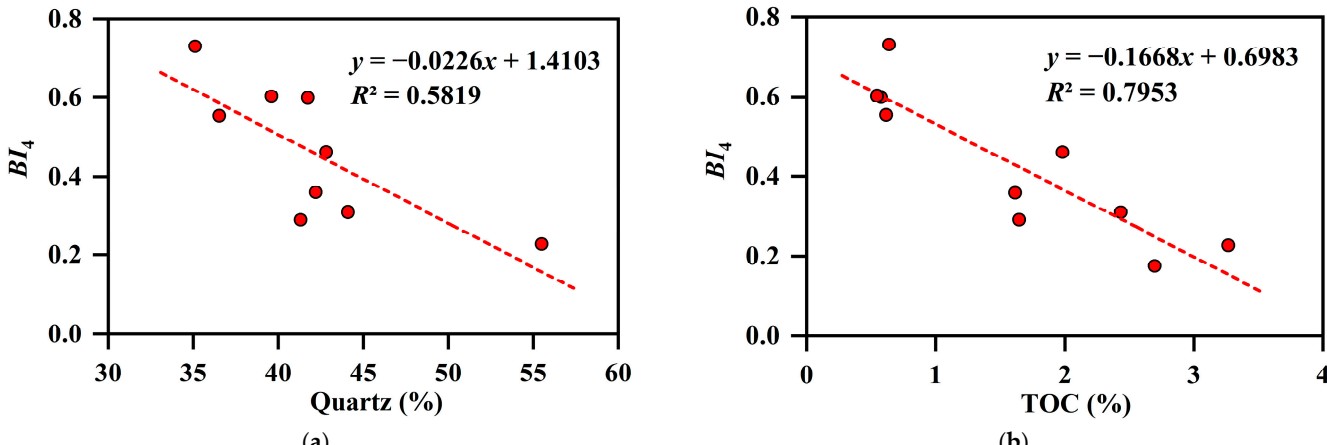

**Figure 7.** Effects of quartz and clay mineral contents on brittleness: (**a**) Effects of quartz on brittleness. (**b**) Effects of clay on brittleness.

*4.6. Shale Brittleness Prediction of Well XQ1*

The rock elastic modulus can be divided into dynamic and static elastic moduli. The Young's modulus calculated using the stress–strain relationship in the rock mechanics experiments was the static modulus. However, the dynamic elastic modulus can be calculated by the propagation velocity of the elastic wave and the density of the rock [51]. Due to the limitations of the rock mechanics experiment, it was difficult to obtain the static elastic modulus. However, the dynamic elastic modulus can be obtained using logging data, and so the static elastic parameters can be estimated by considering the relationship between the dynamic and static elastic parameters to predict the reservoir brittleness characteristics [52].

4.6.1. Calculation of Dynamic Modulus of Elasticity and Conversion of Dynamic and Static Modulus of Elasticity

According to the elastic wave theory, the theoretical relationship between the dynamic elastic modulus of the formation and Poisson's ratio can be calculated using the logging data, as follows [52]:

$$E_d = \rho(3\Delta t_c^2 - 4\Delta t_s^2)/(\Delta t_c^2 \cdot \Delta t_s^2 - \Delta t_s^4), \tag{8}$$

$$\nu_d = (\Delta t_c^2 - 2\Delta t_s^2)/(2\Delta t_c^2 - 2\Delta t_s^2), \tag{9}$$

where $E_d$ is the dynamic Young's modulus, GPa; $\rho$ is the density logging value, g/cm$^3$; $\nu_d$ is the dynamic Poisson's ratio; $\Delta t_s$ is the transverse wave time difference logging value, μs/m; and $\Delta t_c$ is the longitudinal wave time difference logging value, μs/m.

Compared with the experimental static measurement results, the logging data were obtained directly from a wide range of rock media. Therefore, the dynamic elastic modulus and Poisson's ratio of rocks obtained from the logging data are a comprehensive reflection of the rock properties [53]. Ten rock samples from Well XQ1 were taken, and all the samples were measured under the same conditions for the dynamic and static Young's moduli and Poisson's ratio. The measurement results are shown in Figure 8.

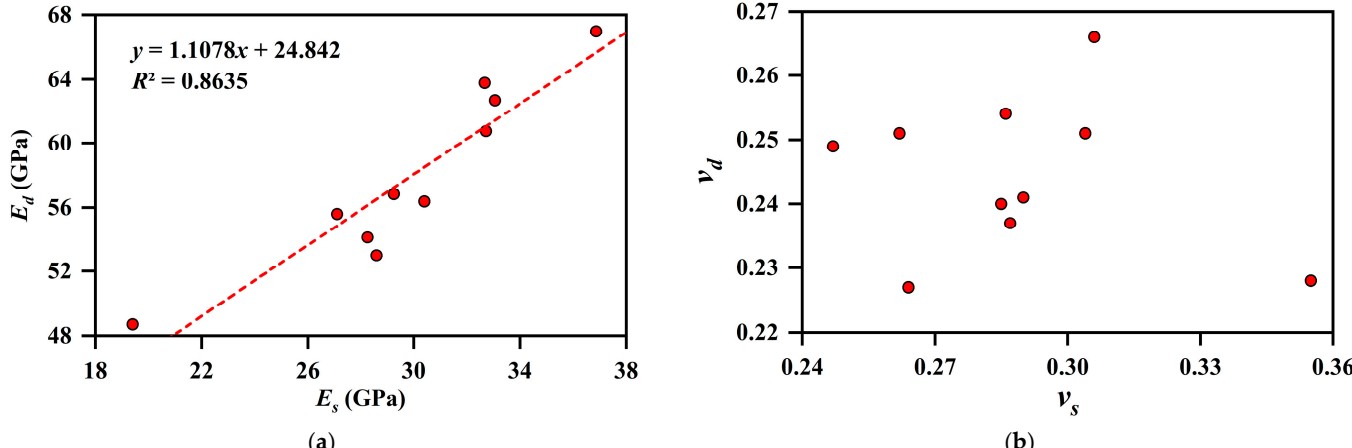

(**a**)　　　　　　　　　　　　　　　　　(**b**)

**Figure 8.** Relationships between dynamic modulus of elasticity and static modulus of elasticity: (**a**) Relationships between dynamic Young's modulus and static Young's modulus. (**b**) Relationships between dynamic Poisson's ratio and static Poisson's ratio.

Figure 8 indicates that the dynamic Young's modulus values calculated under reservoir conditions were higher than the static Young's modulus values. The dynamic and static Young's moduli had an excellent linear correlation with a correlation coefficient of 0.8635. However, the dynamic and static Poisson's ratios had no significant correlation.

### 4.6.2. Hardness Prediction

Continuous rock hardness data could not be obtained with the limited number of core samples. Thus, the logging curves were used to predict rock hardness in this study. Previous studies indicate that neural networks are an effective technology for estimating various parameters of reservoirs or source rocks, such as TOC, porosity, etc. [54]. This study used the BP neural network method to estimate hardness. However, we first determined the optimal logging curves. The AC, CNL, and C13 curves were selected according to the correlations between the logging curves and hardness, as exhibited in Figure 9. The calculated hardness values showed an excellent correlation with measured values (Figure 10), meaning that the BP neural network model established in this study is accurate.

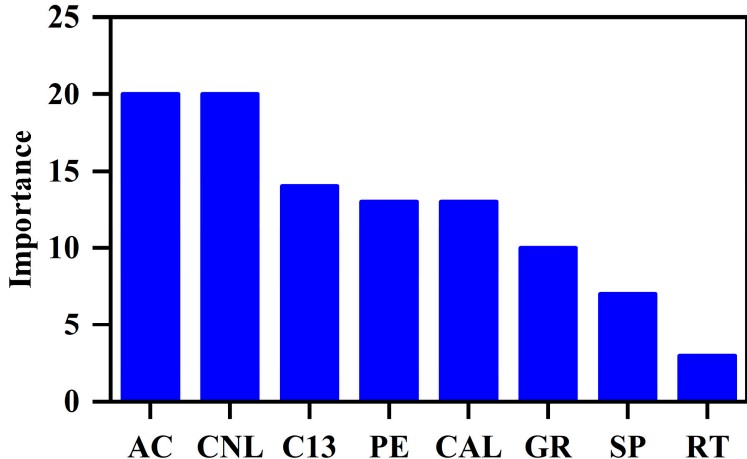

**Figure 9.** Importance of various logging parameters for prediction.

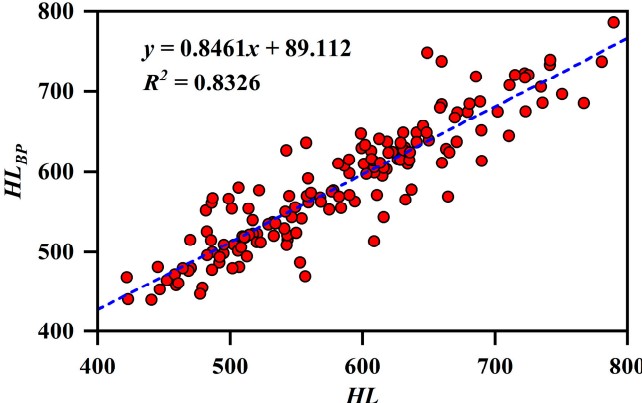

**Figure 10.** Relationship between calculated and measured hardness values. *HL* indicates the actual tested Richter hardness of the studied samples, and *HL*$_{BP}$ shows the hardness predicted using the neural network method.

### 4.6.3. Brittleness Prediction

Based on the dynamic elastic modulus calculated from the logging curves and the rock hardness predicted by the neural network model, the shale brittleness in Well XQ1 was obtained, as shown in Figure 11. For the shales, the Ed was between 52.22 GPa and 64.31 GPa (mean of 59.37 GPa). The HLBP varied from 420.50 to 794.88, with an average of 595.64, and the average value of BI$_{BP}$ was 0.359 (0.09–0.569). The rock hardness and dynamic elastic modulus were significantly correlated, and Young's modulus and Poisson's ratio were also higher for high-hardness rocks. The rock density was poorly correlated with the BI$_{BP}$. Moreover, in Well XQ1, the brittleness of sub-member I of the Lianggaoshan Formation (0.447) was found to be greater than that of II (0.354) and III (0.326), suggesting that sub-member I is more prone to forming complex fracture networks during fracturing.

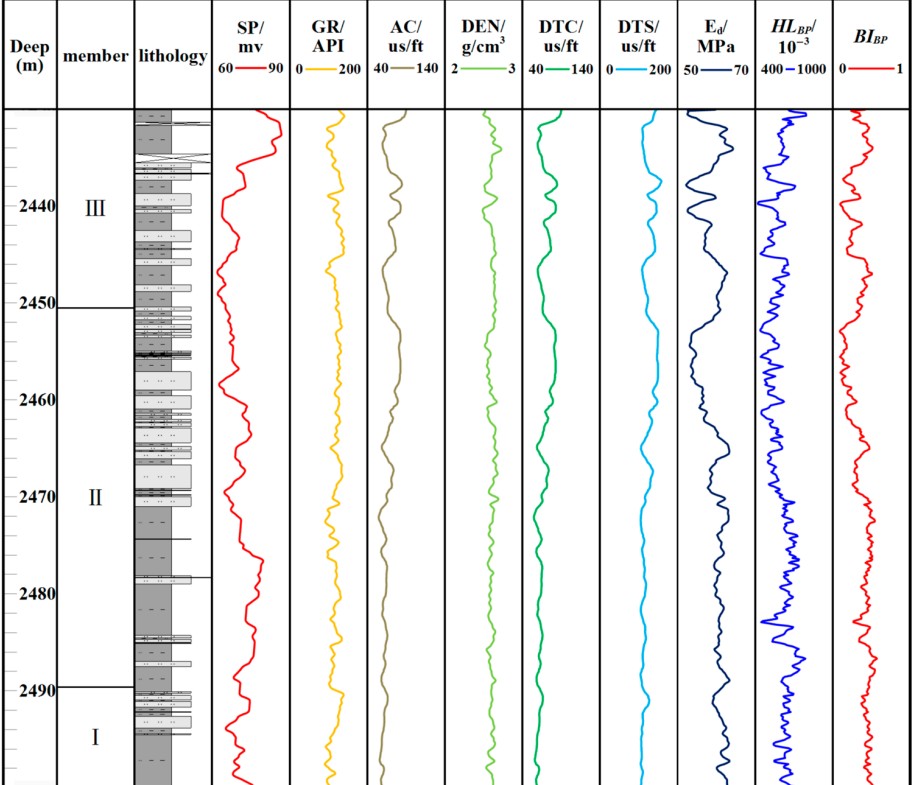

**Figure 11.** Brittleness prediction for Well XQ1 in Lianggaoshan Formation.

## 5. Conclusions

The stress–strain curves of the studied samples mainly showed elastic–plastic deformation, and the damage mode of the test samples was mainly shear damage. For organic-poor shales, the peak stress was larger, and the rock was mainly deformed elastically before rupture. The peak stress of organic-rich shales was smaller.

In this study, $BI_4$, which was derived by calculating the energy change before and after rock damage, was used to fully consider the state change of the shale during the whole process of being fractured. The results of the $BI_4$ calculation are consistent with those of the $BI_1$ calculation with Young's modulus, which shows that $BI_4$ is more suitable than $BI_2$ and $BI_3$.

Shale brittleness is influenced by rock density, hardness, organic matter content, and mineral composition. The more complicated the texture of the sample, the greater the brittleness index. Organic-rich shales have higher toughness and are subject to compaction, and so the organic pores inside the shale are easily compressed and closed, thus reducing the brittleness. However, quartz was proven to negatively influence shale brittleness because the quartz observed in the studied samples was almost all clastic quartz.

The brittleness of different subsections of the XQ1 Well also differed, with the brittleness of sub-member I being higher than that of II and III.

This study clarified the characteristics of shale brittleness and elucidated the controlling factors, the findings of which are beneficial for the development of shale oil in the Sichuan Basin and provide insight into the brittleness of lacustrine shales and its influencing factors, further contributing to the understanding of global petroleum systems.

**Author Contributions:** Conceptualization, H.H. and S.L.; methodology, H.H.; software, H.H.; validation, H.H., P.Z. and Q.Z.; formal analysis, H.H.; investigation, J.W.; resources, J.W.; data curation, Z.L.; writing—original draft preparation, H.H.; writing—review and editing, H.H.; visualization, H.H.; supervision, P.Z.; project administration, S.L.; funding acquisition, S.L. All authors have read and agreed to the published version of the manuscript.

**Funding:** This research was funded by the Natural Science Foundation of Shandong Province (ZR2020QD036). And The APC was funded by S.L.

**Informed Consent Statement:** Informed consent was obtained from all subjects involved in the study.

**Data Availability Statement:** Not applicable.

**Conflicts of Interest:** The authors declare no conflict of interest.

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
