# Peer review of "Evaluation of Lacustrine Shale Brittleness and Its Controlling Factors: A Case Study from the Jurassic Lianggaoshan Formation, Sichuan Basin"

_processes, doi:10.3390/pr11020493_

Round 1
Reviewer 1 Report
The ms is generally well organized with good data to be published.
The following are key concerns for improvement.
1. There are some syntax problems, please revise them. And some can be seen in the attached pdf.
2. Be specific of the four methods in the abstract.
3. Add relative percentages of quartz and clay minerals, which is useful for readers understanding. In abstract.
4. The figure 1 is unclear and please increase the resolution.
5. For the TOC, XRD and SEM measurement, the authors should either add some brief introduction or add necessary citations for the methods used.
6. The scale bar for the figure 2 is unclear, please add necessary explanation.
7.As for the BIs in the Table 5, please add explanation either in the titles or as notes.
8.For the figure 10, the meaning of HL and HLBp should also be specific for readers quick understanding.
9. For Fig. 11, add the units for different columns.
.

Reviewer 2 Report
Please see my attached comments and suggestions, which should be incorporated to make this manuscript acceptable.

Reviewer 3 Report
Please refer to the attachment file. I have some comments for this manuscript.

Round 2
Reviewer 3 Report
No more comments